# The Impact of Atrial Fibrillation on One-Year Mortality in Patients with Severe Lower Extremity Arterial Disease

**DOI:** 10.3390/jcm11071936

**Published:** 2022-03-31

**Authors:** Min-I Su, Ying-Chih Cheng, Yu-Chen Huang, Cheng-Wei Liu

**Affiliations:** 1Division of Cardiology, Department of Internal Medicine, Taitung MacKay Memorial Hospital, Taitung 950, Taiwan; wwfsone@gmail.com; 2MacKay Medical College, New Taipei City 252, Taiwan; 3Graduate Institute of Business Administration, College of Management, National Dong Hwa University, Hualien 974, Taiwan; 4Department of Psychiatry, China Medical University Hsinchu Hospital, China Medical University, Hsinchu 302, Taiwan; b101091022@gmail.com; 5Department of Public Health, Institute of Epidemiology and Preventive Medicine, College of Public Health, National Taiwan University, Taipei 100, Taiwan; 6Research Center of Big Data and Meta-Analysis, Wan Fang Hospital, Taipei Medical University, Taipei 116, Taiwan; dhist2002@yahoo.com.tw; 7Department of Dermatology, Wan Fang Hospital, Taipei Medical University, Taipei 116, Taiwan; 8Department of Dermatology, College of Medicine, Taipei Medical University, Taipei 110, Taiwan; 9Division of Cardiology, Department of Internal Medicine, Tri-Service General Hospital Songshan Branch, National Defense Medical Center, Taipei 105, Taiwan; 10Graduate Institute of Clinical Medicine, College of Medicine, National Taiwan University, Taipei 100, Taiwan

**Keywords:** atrial fibrillation, lower extremity arterial disease, percutaneous transluminal angioplasty, mortality

## Abstract

Atrial fibrillation (Afib) is associated with the presence of lower extremity arterial disease (LEAD), but its effect on a severe LEAD prognosis remains unclear. We investigated the association between Afib and clinical outcomes. We retrospectively enrolled consecutive severe LEAD patients undergoing percutaneous transluminal angioplasty between 1 January 2013 and 31 December 2018. Patients were divided according to the history of any type of Afib and followed for at least one year. The primary outcome was all-cause mortality. Secondary outcomes were cardiac-related mortality and major adverse cardiovascular events (MACEs). The study included 222 patients aged 74 ± 11 years (54% male), and 12.6% had acute limb ischemia. The Afib group had significantly higher rates of all-cause mortality (42.9% vs. 20.1%, *p* = 0.014) and MACEs (32.1% vs. 14.4%, *p* = 0.028) than the non-Afib group. Afib was independently associated with all-cause mortality (adjusted HR: 2.153, 95% CI: 1.084–4.276, *p* = 0.029) and MACEs (adjusted HR: 2.338, 95% CI: 1.054–2.188, *p* = 0.037). The other factors associated with all-cause mortality included acute limb ischemia (adjusted HR: 2.898, 95% CI: 1.504–5.586, *p* = 0.001), Rutherford classification, and heart rate. Afib was significantly associated with increased risks of one-year all-cause mortality and MACEs in patients with severe LEAD. Future studies should investigate whether oral anticoagulants benefit these patients.

## 1. Introduction

Atrial fibrillation and peripheral artery disease (PAD) are common atherosclerotic consequences that interact with each other. Patients with atrial fibrillation have a higher incidence of PAD than those without atrial fibrillation [1]. Conversely, PAD is also associated with an increased risk of developing atrial fibrillation [2]. A nationwide cohort study showed that PAD was independently associated with new-onset atrial fibrillation [3]. One type of severe complications of atrial fibrillation are thromboembolic events, which are frequently diagnosed in PAD patients. These episodes are not limited to cerebrovascular events and include peripheral artery thromboembolic complications, such as splenic, renal, and mesenteric artery infarction, and acute limb ischemia [4]. Overall mortality and cardiac-related mortality were also found to be elevated in patients with atrial fibrillation and symptomatic PAD, independent of age and comorbidities [5]. Lower extremity arterial disease (LEAD), defined by an ankle-brachial index ≤ 0.9, was found to be associated with two-fold higher risks of myocardial infarction and vascular death in patients with atrial fibrillation [6]. Baseline atrial fibrillation was reported to be associated with an increased risk of an incident of acute limb ischemia in a large-scale case-control study [7]. Patients with LEAD without a medical history of atrial fibrillation at the baseline had an increased incidence of atrial fibrillation in a population-based cohort study [8]. Regarding atrial fibrillation and LEAD, it remains unclear which comes first [3,9], but it is known that patients with concomitant atrial fibrillation and PAD have a synergistically elevated risk of cardiac mortality [10]. To date, the effect of atrial fibrillation on prognosis has rarely been reported in patients with severe LEAD. Therefore, we conducted the present study to comprehensively investigate mortality and cardiovascular outcomes in patients with severe LEAD and atrial fibrillation.

## 2. Materials and Methods

We retrospectively enrolled consecutive patients with severe LEAD who received percutaneous transluminal angioplasty at our hospital between 1 January 2013 and 31 December 2018. The present study enrolled all-comers: we excluded three patients with critical limb ischemia and unsalvageable limbs who refused surgical amputation. The patients’ baseline characteristics, laboratory data, procedural details, and outcomes were collected from the medical records. We followed the patients until 31 December 2019. The study was approved by the Mackay Memorial Hospital under Institutional Review Board number 20MMHIS034e. The informed consent was waived by the Mackay Memorial Hospital institutional review board because the present study was a low-risk retrospective cohort study. We conducted the present study in accordance with the Declaration of Helsinki and performed the analyses on anonymized data.

We performed emergency and urgent percutaneous transluminal angioplasty on patients with acute and chronic limb ischemia, respectively; elective percutaneous transluminal angioplasty was reserved for those patients with Rutherford stage IV disease. Physicians could decide whether to use antegrade or retrograde vascular access based on the type of lesion. During the procedure, we routinely administered heparin to maintain an active clotting time between 250 and 300 s and dual antiplatelet agents, namely, aspirin and clopidogrel, according to the current guidelines. Generally, we treated iliac and femoropopliteal lesions with balloon angioplasty, a drug-coated balloon or a stent, and we treated below-knee lesions with only balloon angioplasty. The decision to use a drug-coated balloon or stent was mainly influenced by the patient’s economic status and left to the physicians’ discretion. After the procedure was performed, we continued to administer dual antiplatelet agents for at least one month except in the cases of clinically relevant bleeding complications. Because cilostazol has minimal effects on limb salvage and mortality, we did not routinely prescribe it to severe LEAD patients.

We defined severe LEAD based on the Rutherford classification system as follows: resting pain (stage IV), tissue loss (stage V), and gangrene (stage VI) [11]. The primary study outcome was all-cause mortality at the one-year follow-up. The secondary outcomes were cardiac-related mortality, major adverse cardiac events (MACEs) and major adverse limb events (MALEs) at the one-year follow-up and in-hospital mortality. We defined MACEs as the composite of cardiac-related mortality, nonfatal myocardial infarction, and nonfatal stroke and MALEs as amputation due to a vascular event above the forefoot, acute limb ischemia and clinically driven target vessel revascularization.

We present continuous variables as numbers and standard deviations and binary variables as numbers and percentages. We used independent t-tests and chi-squared tests to evaluate differences in continuous variables and binary variables, respectively. We first used univariate logistic regression analyses to identify potential confounders. Then, we adjusted for these confounders in multivariate logistic regression analyses of the relationships between the presence of atrial fibrillation and the study outcomes. In the statistical models, we adjusted for the presence of acute limb ischemia and the Rutherford classification because these factors obviously contributed to the study outcomes. In the sensitivity analyses, we additionally adjusted for kidney function and body mass index if they were not associated with the study outcomes in multivariate logistic regression. Kidney function was represented by the serum creatinine level, creatinine clearance as determined with the Cockcroft–Gault formula, or the estimated glomerular filtration rate as determined by the Modification of Diet in Renal Disease study equation, as in our previous study [12]. Body mass index was divided by the thresholds of 20 and 25 into groups as follows: <20, ≥20 and <25, and ≥25 [13,14]. We calculated two-tailed *p*-values, and we considered values of 0.05 or lower to be significant. We performed all statistical analyses with the Statistical Package for the Social Sciences software (SPSS software, version 20.0, International Business Machines, Chicago, IL, USA).

## 3. Results

We initially enrolled 225 patients who presented with severe LEAD and excluded three patients who refused surgical amputation. The study cohort consisted of 222 patients aged 74 ± 11 years (54% male), and 12.6% had acute limb ischemia. The patients with atrial fibrillation were significantly older than those without atrial fibrillation (80.6 ± 9.1 vs. 72.6 ± 11.5 years, *p* = 0.001), were more likely to have a history of ischemic stroke (28.6% vs. 13.9%, *p* = 0.056), were more likely to have acute limb ischemia (25.0% vs. 10.8%, *p* = 0.035), had higher CHADS2 scores (2.9 ± 1.3 vs. 2.2 ± 1.2, *p* = 0.004) and higher serum uric acid levels (6.8 ± 2.9 vs. 5.7 ± 2.2 mg/dL, *p* = 0.022). The patients with atrial fibrillation had lower prevalences of chronic kidney disease and end-stage renal disease, and lower serum creatinine levels than those without atrial fibrillation (1.8 ± 1.1 vs. 3.7 ± 3.5 mg/dL, *p* < 0.001). Table 1 describes the details of the baseline characteristics of the study population.

During the study period, the patients with atrial fibrillation had higher incidences of all-cause mortality (42.9% vs. 20.1%, *p* = 0.014), cardiac-related mortality (21.4% vs. 10.3%, *p* = 0.111), MACEs (32.1% vs. 14.4%, *p* = 0.028), MALEs (17.9% vs. 14.9%, *p* = 0.778) and in-hospital mortality (14.3% vs. 6.7%, *p* = 0.242) than those without atrial fibrillation. In univariate logistic regression analyses, atrial fibrillation was significantly associated with increased risks of all-cause mortality (crude hazard ratio (cHR): 2.435, 95% CI: 1.274–4.654, *p* = 0.007) and MACEs (cHR: 2.479, 95% CI: 1.169–5.257, *p* = 0.018). There was a tendency toward a significant association between atrial fibrillation and cardiac-related mortality (cHR: 2.436, 95% CI: 0.977–6.072, *p* = 0.056). No significant associations of atrial fibrillation with MALEs (cHR: 1.280, 95% CI: 0.495–3.307, *p* = 0.610) and in-hospital mortality (cHR: 2.120, 95% CI: 0.691–6.503, *p* = 0.189) were found. Acute limb ischemia and the Rutherford stage were both associated with all-cause mortality (cHR: 4.133, 95% CI: 2.257–7.567, *p* = 0.003 and cHR: 2.073, 95% CI: 1.330–3.233, *p* = 0.001); acute limb ischemia was also associated with cardiac-related mortality (cHR: 3.769, 95% CI: 1.579–8.998, *p* = 0.003) and MACEs (cHR: 2.592, 95% CI: 1.222–5.497, *p* = 0.013), and the Rutherford stage was associated with MALEs (cHR: 4.227, 95% CI: 2.339–7.636, *p* < 0.001). We show the other confounders associated with the study outcomes in Table 2.

In multivariate logistic regression analyses, atrial fibrillation remained significantly associated with all-cause mortality (adjusted HR (aHR): 2.193, 95% CI: 1.109–4.336, *p* = 0.024) and MACEs (aHR: 2.322, 95% CI: 1.045–5.158, *p* = 0.039), independent of acute limb ischemia and Rutherford stage. We did not find a significant association between atrial fibrillation and cardiac-related mortality (aHR: 2.146, 95% CI: 0.819–5.626, *p* = 0.120) or between atrial fibrillation and MALEs (aHR: 1.757, 95% CI: 0.664–4.789, *p* = 0.271) after adjusting for the confounders. Figure 1 shows the Kaplan–Meier curve. Acute limb ischemia (aHR: 2.872, 95% CI: 1.491–5.533, *p* = 0.002) and the Rutherford stage (aHR: 1.918, 95% CI: 1.186–3.103, *p* = 0.008) were significantly associated with all-cause mortality but not cardiac-related mortality or MACEs. The Rutherford stage was significantly associated with MALEs (aHR: 5.577, 95% CI: 2.780–11.19, *p* < 0.001). Heart rate at presentation was significantly associated with all-cause and cardiac-related mortality and MACEs (Table 2).

In the sensitivity analyses, we adjusted for kidney function and body mass index, although these factors were not significantly associated with the study outcomes in univariate logistic regression analyses. The significant association between atrial fibrillation and all-cause mortality remained unchanged after adjustment for kidney function, represented by the creatinine level, creatinine clearance, and the estimated glomerular filtration rate. After we adjusted for body mass index and kidney function in various statistical models, atrial fibrillation was still associated with one-year mortality (aHR: 2.158, 95% CI: 1.012–4.605, *p* = 0.047 for creatinine clearance and aHR: 2.209, 95% CI: 1.036–4.710, *p* = 0.040 for the estimated glomerular filtration rate), but we found only a tendency toward a significant association when the creatinine level was used (aHR: 2.080, 95% CI: 0.950–4.555, *p* = 0.067). Table 3 shows the results of the sensitivity analyses.

## 4. Discussion

In the present study, we showed that in addition to acute limb ischemia and Rutherford stage, atrial fibrillation was an independent predictor of all-cause mortality. The patients with atrial fibrillation had lower creatinine levels but a higher incidence of all-cause mortality than the patients without atrial fibrillation. In the various statistical models, atrial fibrillation doubled the risk of all-cause mortality, and the significant associations were fairly consistent, although only a tendency toward significance was found after we adjusted for body mass index in addition to the other cofounders. In our study, we did not find a significant difference in body mass index between the patients with and without atrial fibrillation, and we therefore did not initially adjust for it as a confounder in our reduced statistical models. The effect of body mass index on mortality remains controversial in patients with LEAD [13,15]. Body mass index was found to have a complex inverse association with the three-year survival rate in patients with critical limb ischemia undergoing endovascular treatment for infrapopliteal lesions in a previous study: overweight patients had the highest one-year survival rate (86.1%), followed by normal-weight patients (78.6%), and underweight patients had the lowest mortality rate (68.4%) [15]. In contrast, the Atherosclerosis Risk in Communities (ARIC) study showed that body mass index was positively associated with the incidence of peripheral artery disease requiring hospitalization, and the risk was relatively higher in patients with critical limb ischemia [13]. We believe that a low body mass index value (<20 kg/m^2^) indirectly reflects a poor nutritional status and compromised wound healing in patients with severe LEAD, whereas a high body mass index value (>25 kg/m^2^) in this population is generally associated with a greater prevalence of cardiovascular comorbidities, such as hypertension, diabetes, and dyslipidemia. Therefore, we suggest that it is beneficial for patients with severe LEAD to maintain a normal weight [13]. The percentage of statin treatment is remarkably low in both groups, particularly in patients with atrial fibrillation (7.1%), whereas treatment with oral anticoagulants was prescribed in only 25% of the latter. Because our hospital is located in a remote area, and most of the patients did not receive regular primary prevention for hypertension, diabetes mellitus, or hyperlipidemia, these patients with the first medical contact with our hospital presented with severe LEAD. Though we tried to increase the prescription rate of statin at discharge, it was not provided by our national health insurance at that time. We believe that guideline-direct medical therapy can improve prognoses in these patients with severe LEAD.

In this study, we enrolled patients with severe LEAD, and all of our patients with atrial fibrillation had CHA2DS2Vsc scores ≥3. According to the current guidelines, oral anticoagulants are strongly recommended for patients with nonvalvular atrial fibrillation and a CHA2DS2Vsc score ≥2 [16,17]. To date, no randomized controlled trial has been published to determine which antithrombotic regimen is best in patients with simultaneous atrial fibrillation and severe LEAD. The ROCKET-AF trial showed that patients with atrial fibrillation and LEAD did not have an elevated risk of stroke or systemic embolization, and a higher risk of major bleeding or nonmajor clinically relevant bleeding was found in patients treated with rivaroxaban than in those treated with warfarin [18]. The ARISTOTLE trial showed that patients with or without LEAD did not have a significantly different risk of stroke or systemic embolization, and the benefits of apixaban and warfarin were similar [19]; however, the two studies enrolled patients with atrial fibrillation and performed a subgroup analysis of patients with LEAD [18,19], which is quite different from a study that enrolled LEAD patients with or without atrial fibrillation, based on the different natures of atrial fibrillation and LEAD. Although these two diseases have concomitant risk factors and share the characteristics of an increased inflammatory burden and a prothrombotic state, endothelial dysfunction and atherosclerosis play major roles in LEAD but not atrial fibrillation [20]. The VOYAGER trial showed that very-low-dose rivaroxaban in addition to aspirin compared with aspirin alone significantly reduced the incidence of acute limb ischemia in patients with peripheral artery disease after revascularization, but all-cause mortality did not differ significantly between the two groups. Notably, the VOYAGER trial excluded patients with atrial fibrillation and patients with major tissue loss [21], but we did not exclude patients with chronic ulcers or major tissue loss of the lower extremities; instead, we enrolled all-comers. The four major randomized-controlled trials included the Afib patients receiving the percutaneous intervention but did not report the medical histories of LEAD in these studies. [22,23,24,25]. A gap existed in patients with concomitant Afib and LEAD, and we provided limited data to complement the studies. Based on the similar pathophysiology of coronary artery disease and LEAD, patients with atrial fibrillation who received percutaneous coronary stenting could be included in our study. The current evidence from four major randomized controlled trials showed that direct oral anticoagulants (DOACs) plus P2Y12 inhibitors can prevent bleeding complications in patients with atrial fibrillation who receive coronary stent(s) without increasing the risk of stent thromboses. Although the selection of the proper antithrombotic regimen for patients with atrial fibrillation and severe LEAD is far beyond the scope of this study, our results showed that atrial fibrillation increased the risk of mortality, and it is worth investigating the effect of antithrombotic treatment in this population in the future.

This study had several limitations. First, we enrolled patients with severe LEAD, so the number of patients in our study population was relatively small. Therefore, we lacked the adequate power to identify a significant association between atrial fibrillation and the other study outcomes after we adjusted for confounders. We did not find a significant association between atrial fibrillation and MACEs, including cardiac-related mortality, nonfatal myocardial infarction, and nonfatal stroke; however, a systematic review and meta-analysis showed that atrial fibrillation was associated with an increased risk of MACEs [26]. Second, our study enrolled all-comers. We noted that a relatively small proportion of the patients had received secondary prevention for dyslipidemia before they presented at our hospital. In addition, the percentage of the patients who had been prescribed oral anticoagulants seemed low in the present study. The percentage of smokers was relatively low in our study, but it is quite unusual in patients with LEAD. We thought some biases might result from a retrospective cohorts study design, such as recall and record biases. Although these biases were non-differential in the two groups, and the statistical results may not change. In our study, 53% of the patients with atrial fibrillation had a creatinine clearance value less than 30 mL/min, and the net clinical benefits should be balanced between antithrombotic treatment and bleeding complications in these patients. We will try to enroll additional patients with atrial fibrillation and prospectively conduct a study to investigate the effect of antithrombotic treatment on the study outcomes.

## 5. Conclusions

Our retrospective cohort study showed that patients with severe LEAD and atrial fibrillation had higher incidences of all-cause and MACEs than those without atrial fibrillation. We showed the adverse effect of atrial fibrillation on all-cause mortality while acute limb ischemia and the Rutherford stage were also associated with all-cause mortality. Future studies should investigate the effect of antithrombotic treatment on mortality in this study population.

## Figures and Tables

**Figure 1 jcm-11-01936-f001:**
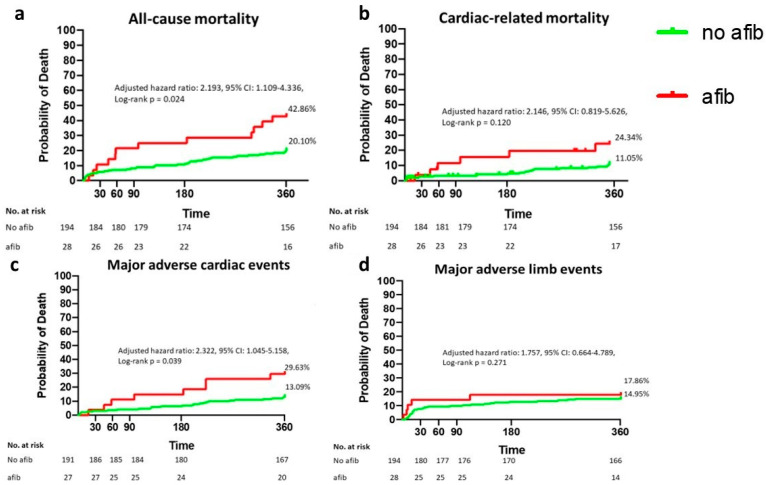
Patients with atrial fibrillation (red line) had significantly greater ratios of all-cause (**a**), cardiac-related mortality (**b**) and major adverse cardiac events (**c**) than patients without atrial fibrillation (green line). Major adverse limb events (**d**) were not significantly different between the two groups.

**Table 1 jcm-11-01936-t001:** Baseline and procedural characteristics and laboratory data in patients with severe lower extremity arterial diseases.

	No Afib		With Afib		*p*
	*n* = 194		*n* = 28		
Age (years)	72.6	(11.5)	80.6	(9.1)	0.001
Male gender	107	55.2%	12	42.9%	0.232
Body mass index (kg/m^2^)	23.8	(4.2)	22.8	(3.9)	0.252
Heart rate at baseline (beats per minute)	87.4	(17.2)	90.7	(19.6)	0.349
Systolic BP at baseline (mm Hg)	148.9	(31.1)	141.8	(27.8)	0.253
Diastolic BP at baseline (mm Hg)	75.6	(14.3)	73.6	(13.1)	0.472
Current/past smoker	49	25.3%	4	14.2%	0.275
Alcohol intake	62	31.9%	7	25.0%	0.678
Family history of premature CAD	2	1%	0	0%	1.000
History of hypertension	124	63.9%	21	75.0%	0.293
History of diabetes mellitus	133	68.6%	17	60.7%	0.398
History of insulin use	27	13.9%	1	3.6%	0.218
History of dyslipidemia	42	21.6%	4	14.3%	0.461
History of kidney disease					0.011
Normal kidney function	117	60.3%	24	85.7%	
Chronic kidney disease	32	16.5%	4	14.3%	
End-stage renal disease	45	23.2%	0	0	
History of CAD	81	41.8%	9	32.1%	0.412
History of myocardial infarction	12	6.2%	2	7.1%	0.692
History of chronic heart failure					0.684
NYHA class I	8	4.1%	2	7.1%	
NYHA class II	10	5.2%	2	7.1%	
NYHA class III	12	6.2%	3	10.7%	
NYHA class IV	5	2.6%	0	0%	
History of carotid artery stenosis	3	1.5%	0	0%	1.000
History of ischemic stroke	27	13.9%	8	28.6%	0.056
History of any cancer	10	5.2%	3	10.7%	0.216
History of amputation	13	6.7%	0	0%	0.949
Above-knee amputation	6	3.1%	0	0%	
Below-knee amputation	3	1.5%	0	0%	
Forefoot amputation	4	2.1%	0	0%	
Presented with acute ischemic limb	21	10.8%	7	25.0%	0.035
Rutherford stages					0.663
Class III	23	11.9%	2	7.1%	
Class IV	45	23.2%	9	32.1%	
Class V	114	58.8%	16	57.1%	
Class VI	12	6.2%	1	3.6%	
CHADS2 score	2.2	(1.2)	2.9	(1.3)	0.004
Laboratory data					
Total cholesterol (mg/dL)	160	(47.0)	148.0	(37.0)	0.204
High-density lipoprotein cholesterol (mg/dL)	40.0	(16.0)	42.0	(27.0)	0.815
Low-density lipoprotein cholesterol (mg/dL)	95.0	(37.0)	93.0	(30.0)	0.902
Triglyceride (mg/dL)	158.0	(125.0)	105.0	(67.0)	0.033
Fasting glucose (mg/dL)	176.0	(99.0)	177.0	(116.0)	0.965
Creatinine (mg/dL)	3.7	(3.5)	1.8	(1.1)	<0.001
Creatinine clearance (mL/min)	34.1	(31.0)	33.3	(19.4)	0.852
Estimated glomerular filtration rate (mL/min/1.73^2^)	41.6	(37.0)	46.2	(28.7)	0.448
Creatinine (mg/dL)	3.7	(3.5)	1.8	(1.1)	<0.001
Alanine transaminase (IU/L)	22.0	(19.0)	30.0	(31.0)	0.150
Uric acid (mg/dL)	5.7	(2.2)	6.8	(2.9)	0.022
White blood cell count 10^3^/µL	9645	(4970)	8593	(4623)	0.029
Neutrophil ratio (%)	70.5	(13.3)	67.7	(17.4)	0.330
Lymphocyte ratio (%)	17.1	(10.1)	17.0	(8.4)	0.969
NLR	7.9	(10.0)	6.3	(6.4)	0.421
Medication use at baseline					
Aspirin	70	36.1%	12	42.9%	0.533
Cilostazol	93	47.9%	15	53.6%	0.687
Clopidogrel	57	29.4%	4	14.3%	0.115
Oral anti-coagulants	0	0%	7	25.0%	<0.001
Pentoxifylline	8	4.1%	3	10.7%	0.148
ACEI or ARB	9	4.6%	2	7.1%	0.634
Beta-blocker	33	17.0%	6	21.4%	0.596
Calcium channel blocker	39	20.1%	9	32.1%	0.149
Statin	42	21.6%	2	7.1%	0.080
Urate lowering therapy	3	1.5%	2	7.1%	0.121

Values are expressed as numbers (standard deviation) or numbers and percentages. Afib = atrial fibrillation; CAD = coronary artery disease; NYHA = New York Heart Association; BP = blood pressure; NLR = neutrophil-to-lymphocyte ratio; ACEI = angiotensin-converting enzyme inhibitor; ARB = angiotensin receptor blocker.

**Table 2 jcm-11-01936-t002:** Variables associated with study outcomes in patients with critical limb ischemia in logistic regression analyses.

	Crude Hazard Ratio	95% CI	*p*	Adjusted Hazard Ratio	95% CI	*p*
All-cause mortality at one year						
Atrial fibrillation	2.435	1.274−4.654	0.007	2.193	1.109−4.336	0.024
Age (years)	1.033	1.006−1.060	0.016	1.018	0.991−1.046	0.200
Male gender	0.678	0.390−1.177	0.167	0.693	0.389−1.235	0.214
Body mass index (kg/m^2^)	0.918	0.850−0.990	0.027			
Smoking	0.925	0.558−1.532	0.762			
Heart rate (beats per minute)	1.026	1.011−1.042	0.001	1.019	1.003−1.034	0.017
Systolic blood pressure (mm Hg)	0.995	0.986−1.005	0.325			
Diastolic blood pressure (mm Hg)	1.013	0.992−1.033	0.226			
Hypertension (yes or no)	0.675	0.388−1.175	0.164			
Dyslipidemia (yes or no)	0.777	0.440−1.370	0.383			
Chronic heart failure (yes or no)	1.021	0.512−2.039	0.952			
Ischemic stroke (yes or no)	1.361	0.681−2.717	0.383			
Acute ischemic limb (yes or no)	4.133	2.257−7.567	<0.001	2.872	1.491−5.533	0.002
Hypertension (yes or no)	0.675	0.388−1.175	0.164			
Rutherford classification (IV~VI)	2.073	1.330−3.233	0.001	1.918	1.186−3.103	0.008
Creatinine (mg/dL)	1.018	0.943−1.099	0.640			
Triglyceride (mg/dL)	0.997	0.993−1.001	0.095			
Uric acid (mg/dL)	1.017	0.888−1.164	0.810			
NLR	1.030	1.016−1.044	<0.001	1.032	1.015−1.050	<0.001
CHADS2 score	1.016	0.804−1.285	0.891			
Cardiac-related mortality at one year						
Atrial fibrillation	2.436	0.977−6.072	0.056	2.146	0.819−5.626	0.120
Age (years)	1.031	0.994−1.069	0.100	1.017	0.980−1.056	0.373
Male gender	0.703	0.325−1.520	0.371	0.712	0.319−1.588	0.407
Body mass index (kg/m^2^)	0.917	0.826−1.016	0.098			
Smoking	1.027	0.525−2.009	0.938			
Heart rate (beats per minute)	1.039	1.018−1.061	<0.001	1.032	1.011−1.053	0.003
Systolic blood pressure (mm Hg)	0.999	0.986−1.013	0.928			
Diastolic blood pressure (mm Hg)	1.021	0.993−1.049	0.151			
Hypertension (yes or no)	0.694	0.319−1.511	0.358			
Dyslipidemia	1.246	0.524−2.963	0.619			
Chronic heart failure (yes or no)	1.551	0.652−3.690	0.321			
Ischemic stroke (yes or no)	0.737	0.221−2.455	0.619			
Acute ischemic limb (yes or no)	3.769	1.579−8.998	0.003	2.247	0.862−5.856	0.098
Rutherford classification (IV~VI)	1.346	0.778−2.327	0.288	1.189	0.668−2.114	0.557
Creatinine (mg/dL)	1.007	0.902−1.125	0.897			
Triglyceride (mg/dL)	0.997	0.993−1.002	0.239			
Uric acid (mg/dL)	1.040	0.860−1.257	0.688			
NLR	1.027	1.006−1.049	0.011	1.031	1.004−1.059	0.023
CHADS2 score	1.029	0.741−1.428	0.867			
Major adverse cardiac events at one year						
Atrial fibrillation	2.479	1.169−5.257	0.018	2.322	1.045−5.158	0.039
Age (years)	1.016	0.986−1.046	0.300	1.003	0.972−1.034	0.866
Male gender	0.805	0.422−1.534	0.510	0.858	0.446−1.649	0.645
Body mass index (kg/m^2^)	0.923	0.849−1.004	0.062			
Smoking	1.523	0.945−2.454	0.084			
Heart rate (beats per minute)	1.025	1.008−1.042	0.004	1.020	1.002−1.037	0.025
Systolic blood pressure (mm Hg)	0.998	0.988−1.009	0.715			
Diastolic blood pressure (mm Hg)	1.008	0.985−1.031	0.506			
Hypertension (yes or no)	0.753	0.391−1.452	0.398			
Dyslipidemia	1.318	0.638−2.723	0.456			
Chronic heart failure (yes or no)	1.654	0.800−3.416	0.174			
Ischemic stroke (yes or no)	1.044	0.436−2.503	0.923			
Acute ischemic limb (yes or no)	2.592	1.222−5.497	0.013	1.609	0.697−3.719	0.265
Rutherford classification (IV~VI)	1.257	0.806−1.960	0.314	1.133	0.711−1.805	0.600
Creatinine (mg/dL)	0.973	0.877−1.080	0.610			
Triglyceride (mg/dL)	0.999	0.995−1.002	0.434			
Uric acid (mg/dL)	1.047	0.905−1.213	0.536			
NLR	1.016	0.993−1.039	0.176	1.016	0.988−1.044	0.277
CHADS2 score	1.095	0.834−1.438	0.513			
Major adverse limb events at one year						
Atrial fibrillation	1.280	0.495−3.307	0.610	1.757	0.664−4.789	0.271
Age (years)	0.979	0.952−1.007	0.146	0.972	0.943−1.002	0.067
Male gender	1.145	0.582−2.253	0.696	1.709	0.851−3.433	0.132
Body mass index (kg/m^2^)	0.941	0.864−1.025	0.160			
Smoking	1.009	0.553−1.841	0.977			
Heart rate (beats per minute)	0.999	0.980−1.018	0.926			
Systolic blood pressure (mm Hg)	0.991	0.980−1.002	0.105			
Diastolic blood pressure (mm Hg)	0.984	0.960−1.008	0.196			
Hypertension (yes or no)	0.842	0.422−1.682	0.627			
Dyslipidemia	1.332	0.622−2.854	0.461			
Chronic heart failure (yes or no)	2.160	1.053−4.433	0.036	2.158	1.025−4.544	0.043
Ischemic stroke (yes or no)	1.151	0.476−2.779	0.755			
Acute ischemic limb (yes or no)	0.976	0.344−2.770	0.963	0.785	0.258−2.385	0.669
Rutherford classification (IV~VI)	4.227	2.339−7.636	<0.001	5.577	2.780−11.19	<0.001
Creatinine (mg/dL)	1.044	0.957−1.138	0.337			
Triglyceride (mg/dL)	1.000	0.998−1.003	0.844			
Uric acid (mg/dL)	0.906	0.765−1.073	0.251			
NLR	1.016	0.992−1.040	0.190	1.002	0.973−1.033	0.876
CHADS2 score	1.091	0.826−1.442	0.539			

CI = confidence interval.

**Table 3 jcm-11-01936-t003:** The association between atrial fibrillation and the study outcomes in the sensitivity analyses.

The Study Outcomes	Adjusted Hazard Ratio	95% CI	*p*
All-cause mortality at one year			
Model 1	2.416	1.204−4.845	0.013
Model 2	2.141	1.047−4.377	0.037
Model 3	2.533	1.304−4.917	0.006
Model 4	2.080	0.950−4.555	0.067
Model 5	2.158	1.012−4.605	0.047
Model 6	2.209	1.036−4.710	0.040
Cardiac-related mortality at one year			
Model 1	2.276	0.853−6.075	0.101
Model 2	2.176	0.801−5.913	0.127
Model 3	2.483	0.979−6.297	0.055
Model 4	2.022	0.708−5.778	0.189
Model 5	2.169	0.789−5.963	0.133
Model 6	2.169	0.792−5.938	0.132
Major adverse cardiac events at one year			
Model 1	2.303	1.032−5.140	0.042
Model 2	1.869	0.795−4.393	0.151
Model 3	2.408	1.117−5.191	0.025
Model 4	1.682	0.699−4.048	0.246
Model 5	1.780	0.760−4.172	0.184
Model 6	1.757	0.753−4.099	0.192

Model 1 adjusted for creatinine in addition to age, gender, heart rate, the presence of acute limb ischemia, Rutherford classification stages, and neutrophil-to-lymphocyte ratio. Model 2 adjusted for creatinine clearance in addition to heart rate, the presence of acute limb ischemia, Rutherford classification stages, and neutrophil-to-lymphocyte ratio. Model 3 adjusted for estimated glomerular filtration rate in addition to heart rate, the presence of acute limb ischemia, Rutherford classification stages, and neutrophil-to-lymphocyte ratio. Model 4 additionally adjusted for body mass index on the top of model 1. Model 5 additionally adjusted for body mass index on the top of model 2. Model 6 additionally adjusted for body mass index on the top of model 3.

## Data Availability

The study data can be provided if the correspondence judges a request is reasonable.

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
