# Peer review of "The Impact of Atrial Fibrillation on One-Year Mortality in Patients with Severe Lower Extremity Arterial Disease"

_jcm, 2022, doi:10.3390/jcm11071936_

Round 1

Reviewer 1 Report

In their study, Su MI et al evaluated the impact of atrial fibrillation on one-year outcomes in patients with severe LEAD undergoing percutaneous revascularization in a single center.

Despite the relatively low number of patients included, mainly due to the highly selected population, the present study is of interest and I believe may add useful insights to the existing literature.

Some minor comments:

  • Introduction, page 2, lines 50-51: LEAD is defined as an ABI ≤ 0.9 (not < 0.9). In cases of incompressible arteries (ABI > 1.40), a toe-brachial index (TBI) ≤ 0.7 is usually used to confirm the diagnosis.
  • Materials and Methods, page 3, lines 104-106: Why did the authors choose BMI and renal function in the sensitivity analysis? I think it is worth justifying the choice. I also suggest adding here some information about the models used (as reported in the caption of Table 3).
  • Results, Table 1: Intriguingly, the percentage of current or former smokers is relatively low in both groups, which is quite unusual in the LEAD population. In contrast, the percentage of diabetic patients is very high. This makes me assume that there is a high percentage of patients with severe LEAD on distal leg disease on a primarily diabetic basis. I do not consider this critical to the study, but was the location of the target lesion evaluated in the analysis?
  • Also in Table 1: the percentage of statin treatment is remarkably low in both groups, particularly in patients with atrial fibrillation (7.1%), whereas treatment with oral anticoagulants was prescribed in only 25% of the latter. This aspect is briefly mentioned in the discussion (limitations), but I think it is worth emphasizing more. How were the patients treated after revascularization? Did they all receive adequate antithrombotic and lipid-lowering treatment, unless contraindicated? I should point out, by the way, that the authors report that "53% of the patients with atrial fibrillation had a creatinine clearance value less than 30 ml/min, and the net clinical benefits should be balanced between antithrombotic treatment and bleeding complications in these patients" (page 9, lines 262-265). However, this aspect is not supported by the results, since normal renal function is reported in 60.3% of patients without atrial fibrillation and in 85.7% of those with atrial fibrillation (see Table 1).
  • Table 2: I believe the Crude hazard ratio in the right hand side is actually the Adjusted hazard ratio.
  • I propose to publish Table 3 as supplemental material. In the caption, please correct "Mode 4" (Model 4).
  • General comment on the discussion: the authors make a long digression on the results of subanalyses of large trials of oral anticoagulants in patients with atrial fibrillation with or without LEAD and the VOYAGER PAD trial. The conclusion is that "it is worth investigating the effect of antithrombotic treatment in this population in the future" (page 9, lines 248-249). However, the study was not designed to evaluate these aspects (as the authors point out). I suggest that the discussion should be oriented more towards the interpretation of the results of the present study. I also recommend strengthening the concluding message. What are the potential implications of these findings? How might they be incorporated into the management of these patients?
  • Discussion, page 9, line 225: Please replace "with and without atrial fibrillation" with "with and without LEAD".
  • Discussion, page 9, lines 239-242: this sentence is unclear to me. Please rephrase.
  • Limitations section, page 9, lines 253-255: again, I do not understand this sentence placed here. Please clarify or remove.

Author Response

In their study, Su MI et al. evaluated the impact of atrial fibrillation on one-year outcomes in patients with severe LEAD undergoing percutaneous revascularization in a single center.

Despite the relatively low number of patients included, mainly due to the highly selected population, the present study is of interest, and I believe may add useful insights to the existing literature.

Some minor comments:

  1. Introduction, page 2, lines 50-51: LEAD is defined as an ABI ≤ 0.9 (not < 0.9). In cases of incompressible arteries (ABI > 1.40), a toe-brachial index (TBI) ≤ 0.7 is usually used to confirm the diagnosis.

Response to the reviewer: Thanks for your comment. We replaced "ABI < 0.9" by "ABI ≤ 0.9". (page line 50-51)

  1. Materials and Methods, page 3, lines 104-106: Why did the authors choose BMI and renal function in the sensitivity analysis? I think it is worth justifying the choice. I also suggest adding here some information about the models used (as reported in the caption of Table 3).

Response to the reviewer: Baseline kidney function or the presence of chronic kidney disease is associated with prognoses in patients with cardiovascular diseases. Therefore, we pre-specified adjusted these factors in sensitivity analyses to evaluate the effect of kidney function on prognoses in the patient with severe LEAD. Similarly, low BMI was associated with poor outcomes in patients with cardiovascular diseases, and we adjusted it in a sensitivity analysis.

  1. Results, Table 1: Intriguingly, the percentage of current or former smokers is relatively low in both groups, which is quite unusual in the LEAD population. In contrast, the percentage of diabetic patients is very high. This makes me assume that there is a high percentage of patients with severe LEAD on distal leg disease on a primarily diabetic basis. I do not consider this critical to the study, but was the location of the target lesion evaluated in the analysis?

Response to the reviewer:

(1) Smoking is a major factor in patients with LEAD. Some biases may occur because we retrospectively enrolled patients, such as recall and record biases. We added a sentence, "The percentage of smokers is relatively low in our study, but it is quite unusual in the patients with LEAD. We thought some biases might result from retrospective cohorts study design, such as recall and record biases. Although these biases were non-differential in the two groups, the statistical results may not change." (line 268-271, page 9).

(2) We had classified the lesion into superficial femoral, popliteal, and below the knee arteries. Still, the location of the target lesion did not play a role in prognoses in our patients with LEAD. We thought that a small number in our study could not differentiate the risk from the various lesion. 

4.Also in Table 1: the percentage of statin treatment is remarkably low in both groups, particularly in patients with atrial fibrillation (7.1%), whereas treatment with oral anticoagulants was prescribed in only 25% of the latter. This aspect is briefly mentioned in the discussion (limitations), but I think it is worth emphasizing more. How were the patients treated after revascularization? Did they all receive adequate antithrombotic and lipid-lowering treatment, unless contraindicated? I should point out, by the way, that the authors report that "53% of the patients with atrial fibrillation had a creatinine clearance value less than 30 ml/min, and the net clinical benefits should be balanced between antithrombotic treatment and bleeding complications in these patients" (page 9, lines 262-265). However, this aspect is not supported by the results, since the normal renal function is reported in 60.3% of patients without atrial fibrillation and in 85.7% of those with atrial fibrillation (see Table 1).

Response to the reviewer: 

We agreed with the reviewer about the lower prescription rate of statin. As we mentioned in our previous study, "A weakness of our study is that we found that the prescription of medication for patients with critical limb ischemia was relatively low at baseline, including the use of statin and angiotensin-converting enzyme inhibitors or angiotensin receptor blockers." (PloS one 16 (5), e0252030). This is because our hospital is located in a remote area, and most of the patients did not receive regular primary prevention for hypertension, diabetes mellitus, or hyperlipidemia—these patients with the first medical contact with our hospital presented with severe LEAD. Though we tried to increase the prescription rate of statin at discharge, it was not provided by our national health insurance at that time. We believe that guideline-direct medical therapy can improve prognoses in these patients with severe LEAD. We added a paragraph on the discussion page 8 lines 215-223.

According to their medical records, we reported the patients' medical histories, such as a history of chronic kidney disease. "Normal renal function is reported in 60.3% of patients without atrial fibrillation and in 85.7% of those with atrial fibrillation in Table 1." But we defined the patients' creatinine clearance value by their serum creatinine level as they presented to our emergent department. In other words, the patient may have a medical history of normal kidney function at baseline but had a creatinine clearance value < 30 ml/min. In table 1, the patients with Afib had mean creatinine clearance values of 33.3 (SD 14.2). We re-checked the ratio of creatinine clearance < 30 ml/min in the two groups, and it is correct that "53% of the patients with atrial fibrillation had a creatinine clearance value less than 30 ml/min."

5. Table 2: I believe the Crude hazard ratio in the right hand side is actually the Adjusted hazard ratio.

Response to the reviewer: We are grateful to you for pointing out this. We apologized for the fundamental error and revised it.

6.I propose to publish Table 3 as supplemental material. In the caption, please correct "Mode 4" (Model 4).

Response to the reviewer: We are grateful to you for pointing out this. We apologized for the fundamental typo and revised it.

7. General comment on the discussion: the authors make a long digression on the results of subanalyses of large trials of oral anticoagulants in patients with atrial fibrillation with or without LEAD and the VOYAGER PAD trial. The conclusion is that "it is worth investigating the effect of antithrombotic treatment in this population in the future" (page 9, lines 248-249). However, the study was not designed to evaluate these aspects (as the authors point out). I suggest that the discussion should be oriented more towards the interpretation of the results of the present study. I also recommend strengthening the concluding message. What are the potential implications of these findings? How might they be incorporated into the management of these patients?

Response to the reviewer: We added three-paragraph (page 8, line 214-222, page 9, line 246-252, and line 270-274, page 10).

"The percentage of statin treatment is remarkably low in both groups, particularly in patients with atrial fibrillation (7.1%), whereas treatment with oral anticoagulants was prescribed in only 25% of the latter. This is because our hospital is located in a remote area. Most of the patients did not receive regular primary prevention for hypertension, diabetes mellitus, or hyperlipidemia—these patients with the first medical contact with our hospital presented with severe LEAD. Though we tried to increase the prescription rate of statin at discharge, it was not provided by our national health insurance at that time. We believe that the guideline-direct medical therapy can improve prognoses in these patients with severe LEAD. (page 8, line 214-222)"

"The four major randomized-controlled trials included the Afib patients receiving the percutaneous intervention but did not report the medical histories of LEAD in these studies.  [22-25]. A gap existed in patients with concomitant Afib and LEAD, and we provided limited data to complement the studies. Based on the similar pathophysiology of coronary artery disease and LEAD, patients with atrial fibrillation who received percutaneous coronary stenting could be included in our study. (page 9, line 246-252)"

"The percentage of smokers is relatively low in our study, but it is quite unusual in the patients with LEAD. We thought some biases might result from retrospective cohorts study design, such as recall and record biases. Although these biases were non-differential in the two groups, the statistical results may not change. (line 270-274, page 10)"

8.Discussion, page 9, line 225: Please replace "with and without atrial fibrillation" with "with and without LEAD."

Response to the reviewer: Thank you again. We revised it as per your suggestion.

9.Discussion, page 9, lines 247-253: this sentence is unclear to me. Please rephrase.

We revised it as, "The four major randomized-controlled trials included the Afib patients receiving the percutaneous intervention but did not report the medical histories of LEAD in these studies. A gap existed in patients with concomitant Afib and LEAD, and we provided limited data to complement the studies. Based on the similar pathophysiology of coronary artery disease and LEAD, patients with atrial fibrillation who received percutaneous coronary stenting could be included in our study."

10.Limitations section, page 9, lines 253-255: again, I do not understand this sentence placed here. Please clarify or remove.

Response to the reviewer: We remove it as the reviewer's suggestion. The removed sentence was, "However, the ROCKET-AF and ARISTOTLE trials had 884 and 839 atrial fibrillation patients with a medical history of LEAD, respectively [18,19]."

Reviewer 2 Report

The authors aim to investigate mortality and cardiovascular outcomes in patients with severe peripheral arterial occlusive disease. The title, however, suggests that the impact of arterial fibrillation on mortality is investigated in patients with LEAD. It is unclear what the exact aim of the article is. The number of patients with Afib in this study is small, yet intensive statistical analysis is performed with high risk of type I or II error. The statistical power of this article is therefore limited, and so are the conclusions.

  1. The authors report that atrial fibrillation is associated with (cardiac-related) mortality and major adverse cardiac events. This is not new and has been reported for much larger groups of patients. The same is true for the mortality risk for acute ischemia compared to chronic ischemia and Rutherford stage. The article suggests that Afib and LEAD originate from the same
  2. Severe LEAD was defined as Rutherford IV – VI, which is a classification used for chronic limb-threatening ischemia (CLTI). As I understand it the patient group is a composite of patients with chronic and acute ischemia. These are different diseases with different causes, and therefore different classifications. (Overview of Classification Systems in Peripheral Artery Disease, Rulon L. Hardman, MD, PhD,1 Omid Jazaeri, MD,1,2 J. Yi, MD,2 M. Smith, MD,1 and Rajan Gupta, MD1). Please clarify.
  3. The statistical analysis is unclear. How was the multivariate regression performed? Which parameters were given as input? What is the risk of type I and type II error with this amount of variables in this number of patients?
  4. How do the authors explain that Afib was associated with higher risk of all-cause mortality, but not with cardiac-related mortality? This is counterintuitive. Could it be that the cause of mortality was underreported?
  5. The conclusion states that cardiac-related mortality was different between the groups, however this was not significant.
  6. Lines 40 and 44: Patients with atrial fibrillation had a higher incidence 44 of PAD is stated twice

Author Response

Reviewer 2

The authors aim to investigate mortality and cardiovascular outcomes in patients with severe peripheral arterial occlusive disease. The title, however, suggests that the impact of arterial fibrillation on mortality is investigated in patients with LEAD. It is unclear what the exact aim of the article is. The number of patients with Afib in this study is small, yet intensive statistical analysis is performed with high risk of type I or II error. The statistical power of this article is therefore limited, and so are the conclusions.

  1. The authors report that atrial fibrillation is associated with (cardiac-related) mortality and major adverse cardiac events. This is not new and has been reported for much larger groups of patients. The same is true for the mortality risk for acute ischemia compared to chronic ischemia and Rutherford stage. The article suggests that Afib and LEAD originate from the same. Severe LEAD was defined as Rutherford IV – VI, which is a classification used for chronic limb-threatening ischemia (CLTI). As I understand it the patient group is a composite of patients with chronic and acute ischemia. These are different diseases with different causes, and therefore different classifications. (Overview of Classification Systems in Peripheral Artery Disease, Rulon L. Hardman, MD, PhD,1 Omid Jazaeri, MD,1,2 J. Yi, MD,2 M. Smith, MD,1 and Rajan Gupta, MD1). Please clarify.

Response to the reviewer: Thanks for the reviewer’s comment. We enrolled the patients with acute limb ischemia in the Rutheford category IIa and IIb, and the three patients with a non-salvageable limb  (the Rutherford category III) were excluded. We added the sentence into the definition in the method to avoid misunderstanding, “ We enrolled the patients with acute limb ischemia in the Rutheford category IIa and IIb, and excluded those patients with a non-salvageable limb  (the Rutherford category III)  (page 2 line 90).” Although Rutheford stages were originally used for LEAD, some of the patients with acute limb ischemia may have histories of LEAD. Therefore, we would keep the Rutheford stages for the enrolled patients, including severe LEAD and acute limb ischemia, in Table 1. Our study aimed to investigate the effect of Afib on prognoses in patients with severe LEAD. We cannot exclude the patients with acute limb ischemia because Afib may play a major role to result in acute limb ischemia in our cohort

  1. The statistical analysis is unclear. How was the multivariate regression performed? Which parameters were given as input? What is the risk of type I and type II error with this amount of variables in this number of patients?

Response to the reviewer: As we mentioned in the method (page 2, line 99-102), “We first used univariate logistic regression analyses to identify potential confounders. Then, we adjusted for these confounders in multivariate logistic regression analyses of the relationships between the presence of atrial fibrillation and the study outcomes.” If variables were significantly associated with the study outcomes in a univariate logistic regression analyses, then we adjusted these confounders in multivariate logistic regression analyses. 

We presented univariate and multivariate logistic regression analyses as crude and adjusted hazard ratios in Table 2. Type I and II errors may exist with respect to cardiac-mortality and the other study outcomes. That was why we addressed the issue of small case number in the present study. As we mentioned in the study limitation, “This study had several limitations. First, we enrolled patients with severe LEAD, so the number of patients in our study population was relatively small. Therefore, we lacked adequate power to identify a significant association between atrial fibrillation and the other study outcomes after we adjusted for confounders (page 9, line 256-259).” Because we only provided limited evidence from our small cohort study, the more number of the patients should be enrolled. As we mentioned, “We will try to enroll additional patients with atrial fibrillation and prospectively conduct a study to investigate the effect of antithrombotic treatment on the study outcomes (page 9 line 274-276)”.

3.How do the authors explain that Afib was associated with higher risk of all-cause mortality, but not with cardiac-related mortality? This is counterintuitive. Could it be that the cause of mortality was underreported?

Response to the reviewer: The event rates in the all-cause and cardiac-related mortality may explain the counterintuitive between the two study outcomes. In other words, type II errors may occur regarding cardiac-related mortality due to the small case number in the present study. “During the study period, the patients with atrial fibrillation had higher incidences of all-cause mortality (42.9% vs. 20.1%, P = 0.014) and cardiac-related mortality (21.4% vs. 10.3%, P = 0.111) (page 4 line 32-34).”

Though we checked the survival from the medical records in the present study, we ascertain the accuracy of the mortality rate in the present study. Notably, some of the LEAD patients were dead of septic shock from soft-tissue infection and we classified it as all-cause mortality. We cannot exclude these patients were dead of reperfusion injury, and it should be classified as cardiac-related mortality. The inconsistency between all-cause and cardiac-related mortality may be partially related to the classification. 

4.The conclusion states that cardiac-related mortality was different between the groups, however this was not significant.

Response to the reviewer: We removed “cardiac-related mortality” in the conclusion to avoid misunderstanding. The revised sentence was as, “Our retrospective cohort study showed that patients with severe LEAD and atrial fibrillation had higher incidences of all-cause and MACEs than those without atrial fibrillation.”

5.Lines 40 and 44: Patients with atrial fibrillation had a higher incidence 44 of PAD is stated twice

Response to the reviewer: Thanks for the reviewer. We deleted the redundancy. 

Round 2

Reviewer 2 Report

The comments have been addressed adequately.